Association of vaccine intention against COVID-19 using the 5C Scale and its constructs: a Pima County, Arizona cross-sectional survey

Block Ngaybe Maiya G. 1 mgblock@arizona.edu
Mantina Namoonga 1
Pope Benjamin 1
Raghuraman Veena 1
Marczak Jacob 1
Velickovic Sonja 1
Jordan Dominique 1
Kinkade Mary 2
http://orcid.org/0000-0002-6067-0861 Perez-Velez Carlos Mario 2 3
http://orcid.org/0000-0002-6411-802X Krauss Beatrice J. 4
Advani Shailesh M. 5
Bell Melanie 1
Madhivanan Purnima 1 3
1 Mel and Enid Zuckerman College of Public Health , Tucson, Arizona , United States
2 Division of Epidemiology, Pima County Health Department , Tucson, Arizona , United States
3 Division of Infectious Diseases, College of Medicine, University of Arizona , Tucson, Arizona , United States
4 Psychology Department, College of Science, University of Arizona , Tucson, Arizona , United States
5 Department of Internal Medicine, Roswell Park Comprehensive Cancer Center , Buffalo, New York , United States
Thorrington Dominic
Electronic publication date: 2024 Dec 6
Publication date: 2024
Volume: 12
Electronic Location ID: e18316
Received 2024 Apr 19; Accepted 2024 Sep 24
Copyright: © 2024 Block Ngaybe et al.
Copyright year: 2024
Copyright holder: Block Ngaybe et al.
License: This is an open access article distributed under the terms of the Creative Commons Attribution License, which permits unrestricted use, distribution, reproduction and adaptation in any medium and for any purpose provided that it is properly attributed. For attribution, the original author(s), title, publication source (PeerJ) and either DOI or URL of the article must be cited.
License URL: https://creativecommons.org/licenses/by/4.0/

Keywords: Vaccination refusal, COVID-19 vaccine, Intentions, Acceptability, Vaccination, Patient acceptance of health care, COVID-19

Funding: NIH/FIC D43 TW010540 Pima County Health Department National Institutes of Health, Fogarty International Center/Global Health Equity Scholars Fellowship D43TW010540 This study was conducted using Purnima Madhivanan’s start-up funds. Purnima Madhivanan is supported by the GHES training grant from NIH/FIC under award number D43 TW010540. The Pima County Health Department provided some funds to aid data collection. The content is solely the responsibility of the authors and does not necessarily represent the official views of the funders. Maiya Block Ngaybe was supported by the National Institutes of Health, Fogarty International Center/Global Health Equity Scholars Fellowship (D43TW010540). There was no additional external funding received for this study. The funders had no role in study design, data collection and analysis, decision to publish, or preparation of the manuscript.

==============================
Background

Vaccine hesitancy has been ranked as one of the top 10 threats to global health by the World Health Organization. The 5C model (Confidence, Calculation of risk, Complacency, Collective Responsibility, and Constraints) and an accompanying tool to measure vaccine hesitancy, summarize several significant explanatory variables, and move beyond the most common explanatory variable, Confidence.

Methods

From January to May 2021, we administered a cross-sectional survey among adults in Pima County, Arizona in collaboration with the local health department to assess psychological antecedents to (i.e., psychological factors that lead to) COVID-19 vaccination using the 5C Scale. Participants were recruited virtually for the survey using multiple recruitment methods. Unadjusted and adjusted hierarchical ordinal logistic regressions were conducted to determine if the 5C variables had an association with intention to vaccinate (or intent to vaccinate) against COVID-19.

Results

Of the 1,823 participants who responded to the survey, 924 (76%) were included in the final analyses. Respondents were White (71%), non-Hispanic (59%), Female (68%), Liberal (37%) and Married (46%). The average age of the participants was 43.9 (±1.3) years. Based on the 5C Scale, Confidence (adjOR:3.64, CI [3.08–4.29]), Collective Responsibility (adjOR:1.94, CI [1.57–2.39]) and Complacency (adjOR:0.64, CI [0.51–0.80]) were significantly associated with intention to vaccinate against COVID-19.

Conclusion

Three of the five 5C variables were associated with the intention to vaccinate, two positively and one negatively. A limitation of the study was that the sample was not weighted to be representative of Pima County. Future research should focus on determining which interventions can bolster Confidence and Collective Responsibility attitudes in communities, while dampening Complacency, to better promote vaccine uptake.

Introduction

Vaccine hesitancy, often defined as the “delay in acceptance or refusal of vaccines despite availability of vaccine services,” has been listed among the top ten threats to global health by the World Health Organization (WHO) (The SAGE Vaccine Hesitancy Working Group, 2014). The COVID-19 pandemic has brought its own challenges in vaccine uptake and acceptance, partly due to the uniquely rapid and politicized circumstances of the vaccine development process. As of 9 August 2023, COVID-19 led to more than 6,954,336 deaths globally and 1,137,057 deaths in the United States (US). Vaccination against COVID-19 has provided immense success to reduce both mortality and morbidity associated with COVID-19 (Johns Hopkins Coronavirus Resource Center, 2023). The Center of Disease Control and Prevention (CDC) and Biden Administration provided sequential guidance to the US through the release of the COVID-19 vaccine, starting first with prioritized high-risk categories to all adults above the age of 18 on 28 February 2021. On 18 June 2022, the age-range for Pfizer and Moderna vaccines expanded to 6 months and above (Centers for Disease Control & Prevention, 2023). However, the uptake of COVID-19 vaccinations varies significantly between countries and even between regions within a country such as the US. As of 26 March 2021, the COVID-19 vaccine primary series uptake was still only 18.08% in Pima County (cjonline, 2024). While higher than the national average of 17.5% in the US population, this was still a lower rate than desired at the time as it was lower than the average in Arizona, 28.55% (Mathieu et al., 2020; cjonline, 2024). Hence, it remains crucial to understand the reasons behind vaccine refusal and uptake in different social groups to gain insights on how best to promote vaccinations in different communities around the US. However, while there are many measures to try to understand intention to vaccinate and hesitancy, there has still not been one standardized measure which encompasses a broad range of psychological reasons for vaccination or non-vaccination.

The Strategic Advisory Group of Experts (SAGE) on Immunizations from the WHO developed a Working Group on Vaccine Hesitancy Determinants Matrix and the 3C model to conceptualize the three main categories into which they grouped vaccine hesitancy determinants: Calculation of Risk, Confidence and Convenience. Since then, many other models and tools for assessing vaccine hesitancy have been proposed, yet none have been universally adopted (Larson et al., 2015; Oduwole et al., 2019; Shapiro et al., 2018). The 5C Scale by Betsch et al. (2018) expanded the 3C model into a measure to assess psychological antecedents to (i.e., psychological factors that lead to) vaccination using five domains: Confidence, Calculation, Complacency, Collective Responsibility, and Constraints (Table 1) (Betsch et al., 2018).

Table 1 5C items from questionnaire on self-reported likelihood, i.e., intention, to vaccinate among Pima County adults.

Confidence I am completely confident that COVID-19 vaccines are safe.	
Complacency Vaccination is not necessary because COVID-19 is not common anymore.	
Constraints Everyday stress prevents me from getting vaccinated against COVID-19.	
Calculation When I think about getting vaccinated for COVID-19, I weigh benefits and risks to make the best decision possible.	
Collective responsibility When everyone is vaccinated for COVID-19, I don’t have to get vaccinated, too.	
Intention How likely is it that you will get a COVID-19 vaccine (again) in the future?	

Confidence-related factors which may impact vaccine acceptance may include issues such as “(i) effectiveness and safety of vaccines, (ii) the system that delivers them, including the reliability and competence of the health services and health professionals, and (iii) the motivations of policymakers who decide on the need of vaccines” (Betsch et al., 2018). Therefore, higher Confidence is expected to be positively associated with higher intent-to-vaccinate. Complacency-related factors may include issues such as a low feeling of threat from infectious diseases, low knowledge, low involvement and awareness on the part of individuals. Higher Complacency would be expected to lead to lower intention to vaccinate. Constraints-related factors include issues such as “physical availability, affordability and willingness-to-pay, geographical accessibility, ability to understand (language and health literacy) and appeal of immunization service” (Betsch et al., 2018). Higher Constraints would therefore lead to lower intention to vaccinate. Calculation-related issues would include extensive engagement in information searches and careful consideration of costs and benefits, possibly due to being more risk-averse (Betsch et al., 2018). Higher Calculation would therefore be expected to lead to lower intention to vaccinate. Finally, Collective Responsibility-related factors would be expected to include an interest in protecting others through vaccinating to help contribute to herd immunity in the community. Higher Collective Responsibility would therefore be expected to correspond with higher intention to vaccinate.

This tool has been tested in multiple countries and has a convenient 5-item scale which has been demonstrated to be concurrently valid and reliable in measuring the psychological antecedents to vaccination (Abd ElHafeez et al., 2021; Betsch et al., 2018). There has been some evidence that the 5C Scale factors are significantly associated with the outcome of vaccination intention for COVID-19 (Dorman et al., 2021). In this study we assessed if the 5C variables were associated with the outcome of intention to vaccinate against COVID-19 among adult Pima County residents. Based on findings from a previous qualitative study conducted by the research group, we hypothesized that Confidence and Collective Responsibility would be more highly correlated with uptake of the COVID-19 vaccine in comparison to other variables of Calculation of Risk, Constraints and Complacency in the 5C Scale (Mantina et al., 2022).

Materials and methods

From June 2020 to December 2021, a mixed methods study was conducted to examine the psychological antecedents to COVID-19 and flu intention to vaccinate in Pima County. This article only presents the findings from the quantitative survey conducted from March through May 2021. The qualitative findings from this study are presented in another article (Mantina et al., 2022). Between January and May 2021, we implemented a cross-sectional sociobehavioral survey including the 5C Scale 5-item questionnaire among adults in Pima County, Arizona. The aim of the survey was to examine intention to vaccinate against COVID-19 and seasonal influenza, and factors associated with intentions. To participate in the study, potential participants had to be over the age of 18 years, have the ability to speak and read English or Spanish, report being a resident of Pima County, and have the ability to undergo the informed consent process. Participants were recruited using stratified sampling methods with the intention of oversampling minorities. The Qualtrics platform, purposive mailed and posted recruitment flyers, and snowball sampling using University and Pima County employee networks aided in achieving the sampling goals. A priori, we aimed to recruit 1,000 participants to ensure we would have a size sufficient to complete analyses with sufficient power while integrating enough covariates. However, upon observing that some participant data were incomplete or of poor quality leading to lower complete observations we could utilize in our analysis and subsequently a threat of low power. We therefore increased the number of participants recruited to ensure we would have enough good quality responses to conduct our planned analyses.

The survey instrument was adapted from the Pima County Health Department survey based on themes that emerged from the qualitative phase of the study where 11 focus group discussions were conducted prior to the administration of the survey (Mantina et al., 2022). Survey questions were then translated into Spanish and tested for understandability and flow before implementation. Although the survey was expanded beyond the 5C Scale, Betsch’s scale was included, adapted to focus just on COVID-19 vaccines. The current article reports only on Betsch’s scale and its association with vaccination intent.

The study protocol was reviewed and approved by the University of Arizona Institutional Review Board (IRB Protocol number: 2007796226). All participants were provided with a copy of the approved informed consent form before taking the online survey and were able to stop the survey at any time. Upon completion of the survey, participants were sent five dollars as compensation for their time and effort.

Measures

Outcome and explanatory variables

The outcome variable for this study was ‘likelihood to vaccinate’ against COVID-19 or vaccinate again if already vaccinated, which we will call self-reported “intention to vaccinate” in this article. The question asked as: “How likely is it that you will get a COVID-19 vaccine (again) in the future?” This inclusive wording of all future COVID-19 vaccinations was used to allow this item to capture all participants, including those who may have already received the vaccine, since eligibility criteria were continually changing during the time when this survey was administered. The variable of intention to vaccinate against COVID-19 was measured on a 5-point Likert scale but later on coded into three categories: Unlikely (including a combination of categories ‘Extremely unlikely’ and ‘Unlikely’), Neither likely nor unlikely, and Likely (including a combination of categories ‘Extremely likely’ and ‘Likely’) (Table 1).

The variables of interest are the five domains listed in the five-item version of the 5C scale, described by Betsch et al. (2020) as the psychological antecedents to vaccination (2020). The 5C Scale domains, Confidence, Calculation of Risk, Complacency, Collective Responsibility, and Constraints, were measured using a validated 5-item tool, adapted from the original version of the items to only concern the COVID-19 vaccine rather than vaccines in general. Each item used a 5-point Likert scale scored Strongly Agree (1) to Strongly Disagree (5). See exact wording of the 5C items in Table 1. All measures except for Collective Responsibility were reverse-coded to be in line with other analyses and to ensure that the higher score was expected to be associated with a higher intention-to-vaccine (Wismans et al., 2021; Betsch et al., 2018).

Covariates

Age, gender, race, ethnicity, self-identified political attitude, income, marital status, and education status have been shown to be associated with intention to vaccinate against seasonal influenza and COVID-19 (Kaiser Family Foundation, 2021; Galarce, Minsky & Viswanath, 2011; Mercadante & Law, 2021). Covariate items included in this statistical analysis were gender, ethnicity, race, marital status, level of education achieved, household income, self-identified political attitude, and age based on the behavioral Risk Factor Surveillance System survey questionnaire (Centers for Disease Control and Prevention, 2020). Age was the only variable included as a continuous variable in the model. The gender variable had three categories with ‘Male’ as the reference category, ‘Female’, and a combination of the three remaining categories which had limited responses ‘Other’, ‘Nonbinary/third gender’, and ‘Prefer not to say.’ Hispanic/Latino ethnicity was categorized as ‘Hispanic’, ‘Non-Hispanic’, and a combined category of ‘Don’t know/Not sure’ and ‘Prefer not to say.’ Race categories included ‘White’, ‘Black/African American’, ‘Asian’, ‘Native Hawaiian/Pacific Islander’ combined with ‘American Indian/Alaska Native’, ‘Other race’, ‘Mixed race’ and ‘Prefer not to say.’ All mixed races were grouped into one category to avoid sparse categories. Education was categorized as ‘Less than college graduate,’ ‘College graduate or more,’ and ‘Prefer not to say’. Marital status was defined as ‘Single,’ ‘Divorced’ combined with ‘Separated’ and ‘Widowed’, ‘Married’ combined with ‘Member of an unmarried couple,’ and ‘Prefer not to say.’ Income was categorized as ‘Less than $25,000,’ ‘$25,000 to less than $50,000,’ ‘$50,000 to less than $75,000,’ ‘$75,000 or more’ and ‘Prefer not to say.’ Self-identified political attitude options included ‘Liberal’, ‘Moderate,’ ‘Conservative,’ and ‘Prefer not to say.’ See the list of actual questions and response options in Appendix A.

Statistical analysis

All observations were deidentified prior to analysis. Two researchers assessed each observation to determine if they had enough integrity to be kept in the final dataset used for analysis based on predetermined criteria including quantity of missing data, and logical consistency between responses (e.g., no intention for first shot but intention for second shot). Further random checks were performed to ensure integrity of responses.

For descriptive statistics, univariate associations were examined using chi-square tests for all categorical demographic variables with the outcome variable of intention to vaccinate against COVID-19. We collapsed certain categories to accommodate small sample size in certain groups and to ensure we have power to assess for significant associations. Stata statistical software version 17 (StataCorp LP, College Station, TX, USA) was used to perform all analyses. Statistical significance was assumed at the 0.05 level.

Ordinal logistic regression analysis was conducted with the outcome of ‘intention to vaccinate’. The first model included the 5C variables as the unadjusted model. Then a hierarchical logistic regression model, with Model 0 only having demographic variables, was constructed. In Model 1, Confidence variable was added to Model 0, as it was shown to be the most influential variable of all 5C variables according to the literature (Betsch et al., 2018; Wismans et al., 2021). In Model 2, the Confidence variable was tested on its own since it was predicted that it would have the strongest relationship with the outcome. Finally, in Model 3, the remaining 5C Scale variables were added for the full adjusted model. Odds ratios (OR) and associated 95% confidence intervals (CI) were calculated to evaluate the strength of effect as well as how much each of the C-statistic changed with the addition of new variables to the model.

A sensitivity analysis was conducted by including all eligible data into the same analyses described above to examine if there were any significant differences in the results when data excluded from the final analysis was added back into the analysis.

Results

Eligibility

Although 1,823 participants started the questionnaire, only 1,219 (67% of total) participants were eligible to complete the survey, 1,194 participants successfully completed the survey (98% of those eligible). Possible reasons for nonparticipation could include survey fatigue or the use of bots to complete data automatically (Google Cloud, 2024). Final analysis included 924 participants (76% of eligible participants) due to missing data issues (see Fig. 1).

Figure 1 Participant flow diagram demonstrating inclusion and exclusion of participants for analysis (n = 641).

Description of the data

The largest representation of our participants was female (68%), non-Hispanic (59%), married (46%), liberal (37%), and not a college graduate (53%). Most participants reported that they intended to get vaccinated (77%). Participants significantly more likely to report their intention to vaccinate were: Asian (87%, p < 0.001), college graduates (83%, p = 0.002), married (81% vs 76%, p = 0.005), and liberal (63%, p < 0.001).

There was similar Hispanic representation (41% compared to 38%) in the sample as compared to that of the Pima County population (Data USA, 2021). The percentage of participants who belonged to Minority race/ethnicities were at least 1.5 times the percentages of the general Pima County population in 2019 due to oversampling: 4% in the survey compared to 3.4% in the general population for African American, 6% compared to 2.8% for Asian, and 2% Native Hawaiian or other Pacific Islander participants. There were no American Indian & Alaska Native participants in the final analysis (Table 2).

Table 2 Demographic characteristics of study participants, January–May 2021, n = 924.

	Total N = 924 (col%)	Unlikely to get vaccinated N = 115 n (row%)	Neither likely nor unlikely N = 96 n (row%)	Likely to get vaccinated N = 713 n (row%)	Chi2 P-value	
Gender						
Male	626 (68)	74 (12)	57 (9)	495 (79)	0.10	
Female	281 (30)	41 (15)	37 (13)	203 (72)		
Nonbinary/Other/PNTS	17 (2)	0 (0)	2 (12)	15 (88)		
Ethnicity						
Non-Hispanic	543 (59)	63 (12)	49 (9)	431 (79)	0.19	
Hispanic	370 (40)	52 (14)	45 (12)	273 (74)		
PNTS	11 (1)	0 (0)	2 (18)	9 (82)		
Race						
White	654 (71)	75 (11)	52 (8)	527 (81)	<0.001**	
Black/AA*	41 (4)	5 (12)	13 (32)	23 (56)		
Asian	52 (6)	2 (4)	5 (10)	45 (87)		
Indigenous*	23 (2)	4 (17)	3 (13)	16 (70)		
Other Race	77 (8)	12 (16)	11 (14)	54 (70)		
Mixed	47 (5)	13 (28)	10 (21)	24 (51)		
PNTS	30 (3)	4 (!3)	2 (7)	24 (80)		
Education						
Not college graduate	485 (53)	77 (16)	57 (12)	351 (72)	0.002**	
College graduate	426 (46)	35 (8)	38 (9)	353 (83)		
PNTS	13 (1)	3 (23)	1 (8)	9 (69)		
Marital Status						
Single	328 (36)	42 (13)	38 (12)	248 (76)	0.005**	
Married/Partnered	427 (46)	44 (10)	37 (9)	346 (81)		
Div/Sep/Wid*	155 (17)	28 (18)	16 (10)	111 (72)		
PNTS	14 (2)	1 (82)	5 (36)	8 (57)		
Income (annual in USD)						
Less than 25,000	309 (33)	44 (14)	35 (11)	230 (74)	0.14	
25,000–49,999	244 (60)	32 (13)	31 (13)	181 (74)		
50–74,999	134 (74)	17 (13)	12 (9)	105 (78)		
>75,000	150 (91)	11 (7)	8 (5)	131 (87)		
PNTS	87 (9)	11 (13)	10 (11)	66 (76)		
Political affiliation						
Liberal	338 (37)	14 (4)	20 (6)	304 (90)	<0.001**	
Moderate	271 (29)	28 (10)	31 (11)	212 (78)		
Conservative	144 (16)	37 (26)	17 (12)	90 (63)		
PNTS	171 (19)	36 (21)	28 (16)	107 (63)		
Age, yrs: mean (±SD)	43.9 (±1.3)	43.0 (±3.5)	38.6 (±3.8)	44.6 (±1.5)		
Notes:

** P < 0.05.

Values shown are n (%) where the percentage is of the row totals.

Abbreviations: n, total number; USD, United States dollars; SD, Standard Deviation.

P-values are from Chi-squared tests, except for age where a T-test was used.

* Indigenous = Native Hawaiian or other Pacific Islander or American Indian or Alaska Native, AA, African American; Div/Sep/Wid, Divorced, Separated, Widowed; PNTS, prefer not to say.

In the hierarchical regression, characteristics that were immutable, such as demographic characteristics, were entered first. Then, based on binary correlations, the significant 5C domains measured by the original 5C scale, with each item representing one C, were entered one at a time in order of binary correlation with intention to vaccinate likelihood. Participants’ Confidence levels were associated with higher odds of intention to vaccinate against COVID-19 (adjOR:3.64, CI [3.08–4.29]) in comparison to other 5C variables. Collective Responsibility appeared to have significant higher odds for intention to vaccinate, and the opposite for Complacency (Collective Responsibility adjOR:1.94, CI [1.57–2.39]; Complacency adjOR:0.64, CI [0.51–0.80]) The last two 5C variables, Constraints and Calculation of Risk, did not appear to have a significant association with vaccination intention in unadjusted and adjustment models (Constraints, adjOR:1.03, CI [0.09–1.22]; Calculation, adjOR:1.10, CI [0.95–1.27]). In addition, self-identified political attitude (liberal, moderate, conservative) was significantly associated with intention to vaccinate in the adjusted model; participants who were conservative and who preferred not to say (PNTS) their political affiliation had lowers odds as compared to liberals in their intention to vaccinate (Conservative adjOR:0.51, CI [0.31–0.82]; PNTS adjOR:0.45, CI [0.29–0.70]). Although age, gender (female), race (black and mixed) and income per year (>$75,000) appeared to have significant association to intention to vaccinate in the hierarchical regression models, none of these variables were associated in the final adjusted model (see Table 3). When comparing the different models in the analysis, the variable with the most explanatory power in the model (increasing the McFadden’s R2 value by 0.19) was Confidence.1

Table 3 Hierarchical ordinal logistic regression models for 5C scale’s correlation to Pima County adults’ intention to vaccinate (n = 924).

	Unadjusted OR (95% CI)	Model 0/Dem OR (95% CI)	Model 1/Conf OR (95% CI)	Model 2/Adjusted* OR (95% CI)	
Primary analysis, n = 924	
Age*	–	1.16 [1.06–1.28]	1.09 [0.98–1.21]	1.02 [0.91–1.13]	
Gender (Male)	–				
Female		0.77 [0.58–1.03]	0.56 [0.41–0.78]	0.74 [0.53–1.04]	
Nonbinary/Other/PNTS**		2.74 [0.97–7.75]	1.33 [0.42–4.21]	1.55 [0.47–5.12]	
Ethnicity (Non-Hisp)	–				
Hispanic		1.12 [0.81–1.55]	1.09 [0.77–1.56]	1.03 [0.71–1.48]	
PNTS**		1.11 [0.31–4.02]	1.18 [0.30–4.62]	0.94 [0.24–3.65]	
Race (White)	–				
Black/AA**		0.52 [0.28–0.96]	0.77 [0.39–1.50]	0.74 [0.36–1.53]	
Asian		1.19 [0.67–2.11]	1.13 [0.60–2.13]	0.93 [0.48–1.78]	
Indigenous**		0.48 [0.22–1.06]	0.54 [0.22–1.34]	0.68 [0.27–1.71]	
Other race		0.75 [0.46–1.21]	0.70 [0.41–1.20]	0.84 [0.48–1.46]	
Mixed		0.32 [0.18–0.58]	0.56 [0.29–1.09]	0.53 [0.27–1.02]	
PNTS**		1.30 [0.55–3.11]	1.31 [0.53–3.25]	1.30 [0.52–3.29]	
Education (Not Grad)	–				
College graduate		1.28 [0.95–1.72]	0.91 [0.66–1.27]	0.93 [0.66–1.30]	
PNTS**		0.70 [0.21–2.32]	0.53 [0.15–1.84]	1.15 [0.33–4.02]	
Marital status (Single)	–				
Married/Partnered		0.96 [0.69–1.35]	0.99 [0.68–1.43]	1.15 [0.78–1.69]	
Div/Sep/Wid**		0.79 [0.50–1.25]	0.87 [0.53–1.44]	0.93 [0.56–1.55]	
PNTS**		0.35 [0.12–1.07]	0.54 [0.17–1.73]	0.62 [0.19–1.95]	
Income (<25K USD/yr)	–				
25,000–49,999		0.98 [0.68–1.40]	0.80 [0.54–1.19]	0.86 [0.57–1.29]	
50–74,999		0.92 [0.58–1.47]	0.94 [0.57–1.54]	0.81 [0.49–1.35]	
>75,000		1.86 [1.12–3.08]	1.78 [1.03–3.08]	1.61 [0.91–2.84]	
PNTS**		1.23 [0.73–2.06]	0.82 [0.47–1.46]	0.92 [0.51–1.68]	
Political aff. (Liberal)	–				
Moderate		0.39 [0.27–0.55]	0.56 [0.39–0.83]	0.75 [0.50–1.12]	
Conservative		0.25 [0.17–0.37]	0.36 [0.23–0.55]	0.45 [0.29–0.70]	
PNTS**		0.16 [0.11–0.24]	0.34 [0.21–0.54]	0.51 [0.31–0.82]	
Confidence	3.72 [3.17–4.36]	–	4.24 [3.63–4.96]	3.64 [3.08–4.29]	
Coll responsibility	1.92 [1.57–2.35]	–	–	1.94 [1.57–2.39]	
Complacency	0.60 [0.49–0.74]	–	–	0.64 [0.51–0.80]	
Constraints	0.97 [0.82–1.14]	–	–	1.03 [0.86–1.23]	
Calc of risk	1.12 [0.97–1.29]	–	–	1.10 [0.95–1.27]	
Pseudo R2	0.30	0.07	0.26	0.31	
Notes:

Bolded text means that the p-value was less than 0.05.

Abbreviations: OR, odds ratios; 95% CI, 95% Confidence intervals

Reference groups for categorical categories are single for marital status, less than college graduate for education, and liberal for political status.

* Age ORs are for every 10 years.

** Indigenous = Native Hawaiian or other Pacific Islander or American Indian or Alaska Native, AA, African American; Div/Sep/Wid, Divorced, Separated, Widowed; PNTS, Prefer not to say.

Sensitivity analysis

In the sensitivity analysis, when all eligible and previously excluded observations were included (n = 1,194), the hierarchical ordinal logistic regression demonstrated similar results for the 5C, political affiliation, income, and ethnicity variables (see Appendix B).

Discussion

In our study, Confidence and Collective Responsibility were significantly positively associated with self-reported intention to vaccinate in the future against COVID-19, while Complacency was significantly negatively associated with the intention to vaccinate against COVID-19. Calculation and Constraints did not have an association with intention to vaccinate. These findings are in line with other studies (Kwok et al., 2021; Machida et al., 2021; Wismans et al., 2021). Confidence has been shown to be one of the most salient factors in determining if a population will consider getting vaccinated (Larson, 2018; Nwachukwu et al., 2024). For example, Confidence in vaccines during the COVID-19 pandemic may have been negatively impacted by misinformation and conspiracy theories which may have increased perceptions of vaccine-related risks and lowered intention to vaccinate (Betsch et al., 2018, Mantina et al., 2022). A qualitative study among Latino adults in Arizona found that trust in science and doctors, fear of sickness, a desire to return to life before the pandemic, a perception of vaccination being a civic duty were likely to lead to vaccine Confidence among other factors (Mercado et al., 2024).

Collective Responsibility is known to correlate with collectivism, communal orientation and empathy and negatively with individualism (Betsch et al., 2018). It may have had a higher impact on vaccination intention among our participants since there was a motivation to protect others through vaccination among our participants in the qualitative phase of this study (Betsch et al., 2018; Mantina et al., 2022). Our findings related to Confidence and Collective Responsibility are echoed in a systematic literature review of COVID-19 vaccine hesitancy studies in the US where Confidence was the most prominent characteristic related to vaccination followed by Collective Responsibility (Nwachukwu et al., 2024). However, in this review, Calculation was the next most influential variable on intention to vaccinate, unlike this study in Pima County which found that Complacency was the next most influential of the 5Cs after Confidence and Collective Responsibility (Nwachukwu et al., 2024). This indicates that Complacency may play a bigger role in predicting intention to vaccinate in Pima County in comparison to the rest of the USA.

Complacency occurs when a person does not feel a disease is threatening enough to cause a change in their behavior (Betsch et al., 2018). It may also be related to a feeling of invulnerability and preference for risk-seeking behaviors, and negatively to higher risk perception (Betsch et al., 2018). Another survey in rural parts of Southern Arizona examined willingness to recommend the COVID-19 vaccination to children and found that concern with getting sick with COVID-19 in the future (i.e., higher risk perception) was significantly associated with willingness to recommend the vaccine (Darisi et al., 2023). Our previous qualitative study findings support this study findings on Complacency, where some participants mentioned feeling that they were “healthy” and may not have felt at risk of getting COVID-19 (Mantina et al., 2022). This provides evidence that Complacency may have been a more relevant factor in regard to intention to vaccinate for populations in Southern Arizona compared to other parts of America.

Findings from this study can be used to help inform vaccination promotion efforts, considering which concepts messaging may want to target to be most effective in inspiring intention to vaccinate. One example of a future intervention which could be effective when trying to build confidence in Pima County may include targeted messaging using key opinion leaders at the forefront of the messaging, matched in race/ethnicity and age, or trusted messengers such as recognized doctors from the community. It may also be helpful to promote the fact that very robust data from clinical trials and post-marketing surveillance demonstrate that serious adverse events are very rare and explain how safe and effective the vaccine are (Chou et al., 2020).

Future investigations could collect location-specific data to focus more on how different districts and neighborhoods may differ in their vaccination attitudes and intentions.

Since the present study was conceptualized, there has been another more comprehensive model than the 5C Scale called the 7Cs of Vaccination Readiness which includes two additional important variables Compliance and Conspiracy (Geiger et al., 2022). While this was not used in this study since it was not available at the time, it appears that this model may account for variation which this study could not capture with its limited scope. Accordingly, this would be a good model to use in future studies of this nature. Additionally, the outcome for this study is self-reported intention which may be a predictor of vaccination behavior, although it is not equivalent of the actual behavior. Therefore, future research will need to explore the strength of the relationship of intention to actual vaccination behavior. Economics-based methods such as stated or revealed preference methods may be helpful in narrowing in on specific practical factors which may impact decision-making when it comes to choosing to get a COVID-19 vaccine in the future. It would also be important to conduct more longitudinal studies to monitor attitudes, intentions and forthcoming vaccination behavior over time in Pima County. Studies of this nature could aid in predicting vaccine behavior trends based on psychological characteristics. Ongoing attitudinal and behavioral surveillance may additionally help in catering promotional efforts to specific localities and demographic subpopulations in a more targeted and effective manner.

A limitation of the study is that the sampling methods relied upon convenience and snowball sampling, leading to some possible bias in the sample. As the research team tended to be at the more pro-vaccine end of the spectrum, the networks which may have been accessed in the convenience sampling may have tended towards higher intention to vaccinate in the sample. This could have possibly skewed the data towards an overestimated effect size for Confidence and Collective Responsibility associations with the outcome. The sample is also limited to Pima County which may limit the findings’ generalizability. As with all survey data, there is also risk of recall and desirability bias in the results, probably tending towards a more positive report of vaccination likelihood, i.e., intention to vaccinate.

A strength of the study is that it oversampled for minority populations and is therefore potentially more representative of communities of color than other study samples. Another strength of the study is that it is an analysis of findings from a key period in the pandemic, adding to the understanding of attitudes which may be found with a novel vaccine, and may be able to inform future pandemic responses in Pima County. As pandemics are dynamic, it is critical to have information about the attitudes and intentions of communities in a state of emergency and how they might react to new interventions such as vaccines. We hope that the information presented in this article may help to add to the growing literature on the COVID-19 pandemic and help build a greater understanding of how to prepare and increase acceptability of key interventions such as vaccines more quickly in the future.

Conclusions

In conclusion, in this study we hypothesized that Confidence and Collective Responsibility would be the most highly correlated variables with the outcome of intention to vaccinate against COVID-19. The sensitivity analysis generally supports the conclusions of the primary and secondary analyses. Our hypothesis was confirmed, though a surprising finding was that Complacency was also found to be significantly associated with intention to vaccinate against COVID-19. Future investigations could focus more on how different districts and neighborhoods within Pima County differ in their vaccination attitudes and intentions. The outcome for this study is self-reported intention which may be a predictor, but does not equal future vaccination behavior, so future research will need to explore the strength of the relationship of intention to actual vaccination behavior.

Supplemental Information

Supplemental Information 1 Code for the analysis conducted.

Sections of the analysis contain subheadings within the dofile to aid with understanding how the analysis was conducted.

Supplemental Information 2 The 5C Scale dataset.

The data is described in the data dictionary.

Supplemental Information 3 Data dictionary for the 5C Scale dataset.

This metadata describes the types of information in the dataset.

Supplemental Information 4 STROBE Checklist.

Supplemental Information 5 Questions from questionnaire in 5C Scale Study.

Supplemental Information 6 Sensitivity analysis including all eligible observations – a hierarchical ordinal logistic regression models for 5C scale’s correlation to Pima County adults’ intention to vaccinate (n = 1,194).

We would like to acknowledge the Alliance for Vaccine Literacy research team for their contribution to this research project.

Additional Information and Declarations

Competing Interests

Author Contributions

Ethics

Data Availability

1 While the McFadden’s R2 value is a pseudo R2 value, and therefore cannot truly represent the explanatory power of the model, a larger value indicates a better fit, and a value of 0.2–0.4 is said to be an “excellent fit” (McFadden, 1977).

Carlos M. Perez-Velez and Mary Kinkade were working for the Pima County Health Department during this study. Purnima Madhivanan is faculty at the University of Arizona College of Public Health. The other authors declare that they have no competing interests.

Maiya G. Block Ngaybe conceived and designed the experiments, performed the experiments, analyzed the data, prepared figures and/or tables, authored or reviewed drafts of the article, and approved the final draft.

Namoonga Mantina conceived and designed the experiments, performed the experiments, prepared figures and/or tables, and approved the final draft.

Benjamin Pope conceived and designed the experiments, performed the experiments, prepared figures and/or tables, and approved the final draft.

Veena Raghuraman performed the experiments, prepared figures and/or tables, and approved the final draft.

Jacob Marczak conceived and designed the experiments, performed the experiments, prepared figures and/or tables, and approved the final draft.

Sonja Velickovic performed the experiments, prepared figures and/or tables, and approved the final draft.

Dominique Jordan performed the experiments, prepared figures and/or tables, and approved the final draft.

Mary Kinkade conceived and designed the experiments, prepared figures and/or tables, and approved the final draft.

Carlos Mario Perez-Velez conceived and designed the experiments, prepared figures and/or tables, authored or reviewed drafts of the article, and approved the final draft.

Beatrice J. Krauss conceived and designed the experiments, prepared figures and/or tables, authored or reviewed drafts of the article, and approved the final draft.

Shailesh M. Advani conceived and designed the experiments, prepared figures and/or tables, authored or reviewed drafts of the article, and approved the final draft.

Melanie Bell analyzed the data, prepared figures and/or tables, authored or reviewed drafts of the article, and approved the final draft.

Purnima Madhivanan conceived and designed the experiments, analyzed the data, prepared figures and/or tables, authored or reviewed drafts of the article, and approved the final draft.

The following information was supplied relating to ethical approvals (i.e., approving body and any reference numbers):

All procedures involving human participants were carried out in accordance with the ethical standards of the Institutional Review Board at University of Arizona (IRB Protocol number: 2007796226).

The following information was supplied regarding data availability:

The data, data dictionary, and the STATA code are available in the Supplementary Files.

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
