# Peer review of "Association of vaccine intention against COVID-19 using the 5C Scale and its constructs: a Pima County, Arizona cross-sectional survey"

_PeerJ, doi:10.7717/peerj.18316_

## Round 0.1 · original submission · Minor Revisions

We have received the comments of three reviewers. They are recommending minor revisions to the manuscript.

They have highlighted several aspects of the text that need clarification for the reader, either to explain concepts that won't be familiar to the general reader or to use more precise language. There is an interesting question on why the 5C model was chosen over the 7C model, which the reviewer recommends is addressed in the main text of the manuscript rather than the response to the reviewers.

Please read all of the comments from all three reviewers.

Reviewer 1 ·

Basic reporting

Dear authors,

Thank you for the opportunity to review your manuscript on the 5C's and intention to vaccinate against COVID-19. Overall, it is clearly written and some area's may still need clarification or restructuring. I have provided some comments and suggestions and hope you find these helpful.

Introduction:

When introducing the 5C scale, could you elaborate a bit more on what the different C’s are about? And, maybe also, what is meant with this term ‘antecedents to vaccination’. To readers it may remain a bit abstract what the 5C’s are. As the introduction is a bit short, you could add a paragraph and describe this to provide more context.

Line 73-73; it would be nice if you could clarify for readers what the vaccination policy in the US is. Is COVID-19 (bivalent?) vaccination recommended and offered to everyone? Or only to high-risk groups?

Line 77: I would suggest to replace ‘vaccine intention’ for ‘intention to vaccinate’, as this terminology is more common.

Line 96-100: you could also move the comparison with your prior qualitative study to the discussion section and elaborate a bit more on the comparison.


Conclusion:

332: I would suggest to rephrase ‘not a perfect measure of actual vaccination in the future’ into for example ‘intention may be a predictor, but does not equal future vaccination behaviour’ since it is a bit unclear what is meant. Also adjust accordingly in line 312-313.

334: do you mean longitudinal studies to monitor attitudes, intentions and forthcoming vaccination behaviour over time?

Line 333-338: ‘It… effective manner.’ This is very long sentence and therefore difficult to understand. I would suggest to divide into several short sentences.

General remark on the discussion and conclusion:

There seems some overlap in sentences between the last paragraph of the discussion and conclusion. I would avoid repeating (parts of) sentences. Instead I would suggest to remove or rephrase.

Experimental design

Methods:

Line 103: could you clarify which ‘mixed methods’ are used in your study? This term is mostly used for studies with a quantitative and qualitative component.

Lines 121-128: I would describe here the adaptations and scale that you report on in this study and mention that this was part of a larger questionnaire/project. Only mention the details relevant to this study.

Lines 147-149: here you mention the outcome of interest ‘intention to vaccinate’, which is further explained under the next subheading. I would suggest to merge this information. As this is your main outcome, I would put this at the top.

Line 138: the 5C model is described here as an ‘exposure’. I’m not sure If this is the correct term. You could for example describe ‘intention to vaccinate’ as you primary outcome and the 5C’s as your secondary outcome. Your covariates then describe sociodemographic characteristics.

Results
Line 214: please also refer (explicit) in the materials and methods section what the eligibility criteria were.

Line 218-222: ‘Two of our researchers .. , etc.’ This sentence should be part of the materials and methods section as well ass the following sentences: ‘All observations were .. ensure integrity of the responses. (Line 223-226). Only report here what you found, how many were excluded and why.
Please also refer in the first paragraph to figure 1 (the flow chart).

Validity of the findings

Discussion

Could you elaborate a bit more on how confidence an responsibility are associated with a positive intention to vaccinate and connect this more to the available literature on these topics? It would be interesting to read how this links to qualitative findings of the prior qualitative work that is mentioned. Some examples could also help to make the C’s it for readers less abstracts and easier to understand what is meant.

Line 290: could you explain how you expect that the chosen sampling strategy has biased your results?

Line 289: I would suggest to make the paragraph on limitations / strengths the last one, before your conclusion.

Additional comments

Title of the manuscript:
Due to the cross-sectional nature of you study, 'association' would be more appropriate than 'predicition' which suggests an element of follow-up in time. In your study aim (Line 95) and the discussion (Line 280) you also use the word 'association'. I would also use the same wording in your title.

Abstract: contains a background, methods and results section. This usually also includes a conclusion.

Reviewer 2 ·

Basic reporting

Language used is clear and professional throughout, and the submission format follows the standard. Sufficient background is provided, although I was missing the comparison of the vaccine uptake rate in Arizona and especially in Pima County to the rest of the US. Was it especially low there? Were there any theories on why it was lower than in other regions? Further, in your findings we see a difference in vaccine uptake due to ethnicity and political views, this could also be an interesting point to put into the introductory part to get a better understanding of the situation.

Figure 1 is in low quality and might be uploaded in higher resolution.
Regarding supplemental file 2, I would propose to delete the columns with IP adresses and location data due to privacy reasons.

Minor remarks:
line 64: COVID-9 (instead of COVID-19)
In some lines, there is double spacing or indented text (examples: lines 72, 108, 291).

Experimental design

Why did you not use the 7C model instead of the 5C model? It adds compliance and conspiracy as additional dimension which would be interesting in regard to your findings.
(ex. https://doi.org/10.1027/1015-5759/a000663). I would at least address this in the intro or the method section, even if it was not available at the time of your study.

Minor remarks: add one short explanation of the Qualitrics platform for the uninitiated; what does "some participant data were incomplete or of poor quality" (line 118) mean exactly that it lead you to change the recruitment process?

Validity of the findings

I would like to see the discussion and conclusion section to go more in depth. Your finding of Complacency as a factor is not addressed and put in context. While in your results, constraints did not play a role in vaccine uptake, other studies see this as an influencing factor. I would at least address this and discuss it shortly. (Rancher, Caitlin, et al. "Using the 5C model to understand COVID-19 vaccine hesitancy across a National and South Carolina sample." Journal of Psychiatric Research 160 (2023): 180-186.) Further, while I generally agree with you on your ideas in 301-306, since you emphasize confidence in the government (line 284), is it really enough to just work on this regional/communal level? This is where the ideas from the 7C model would be interesting to discuss.

·

Basic reporting

Overall, the paper is well-written, is coherent and easy to understand. The paper clearly states the hypothesis it intends to test. Some specific suggestions:
Lines 71 – 74: The authors are requested to please update the data on COVID-19 vaccination rates.
Line 96 – If the mentioned qualitative study is published, would be good to reference it here.

Experimental design

Was data collected on respondents’ profession? It would be interesting to see intention of vaccine uptake among priority populations of interest such as physicians, nurses, etc. and how their intention to get vaccinated compared to that of the general population.

Validity of the findings

Lines 284 – 286: The authors say “confidence, especially in government, has been shown to be one of the salient features…..”. Confidence, as defined in the 5C scale (shown in table 1 and in survey tool with variable name "attitudecvd1") is confidence in the safety of the vaccine. The analysis is not sufficient to make any conclusions on the respondents’ confidence on government and its association with their intention to vaccinate. It is advisable to revise the statement to better represent the findings of the analysis.

The discussion can be further enriched by comparing and contrasting existing literature that have used the 3C / 5C model to study COVID-19 vaccination intention. Some discussion on what initiatives were undertaken in Pima county Arizona to improve vaccine uptake and how they were similar or different to approaches addressing the required “confidence” and “collective responsibility” domains this analysis finds would be effective.

Some discussion on existing literature on uptake of other adult vaccinations (eg influenza) in Arizona, and how it differs from COVID-19 vaccine uptake, and the reasons, if any for this difference will enrich the conversation in this paper.

Additional comments

As its been some time since the data was collected, given the dynamic nature of the COVID-19 pandemic and its impact on changing risk perception and potentially intentions for vaccination, the authors may want to note in the discussion the importance of interpreting these results in the context of the time when the data was collected and the limited relevance of these results to a similar stage of a health emergency.

---

## Round 0.2 · Minor Revisions

Thank you revising the manuscript. The reviewers have suggested some very minor additional changes are still required.

Please read their comments carefully to ensure that each comment is taken into account.

Reviewer 1 ·

Basic reporting

Thanks to the authors for their response and explanation.
Regarding my comment on the terminology for 'intention to vaccinate'; I may not have phrased this clearly. What I intended to write is that 'intention to vaccinate' is most commonly used terminology.
The other comments and suggestions have been sufficiently addressed.

In the added paragraphs on the 5C's in the discussion section, it is not entirely clear what is meant with the sentence (line 334-336) 'However, in this review, ... the next most influential of the 5C's'. Please clarify this, and also in relation to previous sentences.

Experimental design

Thanks to the authors for their response and explanation.
I'm not sure if 'explanatory variables' would be an appropriate description of the 5C's as this implies a time-factor which is not studied in a cross-sectional design. Therefore, I would suggest using 'variable' and describe the relationship with intention as an association (as you have already).

Validity of the findings

The addition of how the C's are linked with intention to vaccinate have improved the discussion of your findings. I have not further comments.

Additional comments

No further comments.

Reviewer 2 ·

Basic reporting

Thank you for the revised manuscript. The authors have addressed some of the feedback given by the reviewers, which helps strengthen their findings and places them into more context, making the article more comprehensible and valuable. The background, especially vaccination guidelines and statistics, as well as the different aspects of the 5C Model have been elaborated on in an understandable manner.

I would suggest some minor corrections (minor spelling errors, ex. Participants' Confidence in l. 283; five point Likert (l. 177) / 5-point Likert (l. 187), coherent use of capitalization for the 5Cs), also I think that Reviewer 1 suggested using "intention to vaccinate" throughout instead of using "vaccine intention", as "intention to vaccinate" seems much more commonly used.

Experimental design

It is now clearer that this article reports the quantitative findings of a mixed methods study. The revised Methods Section is sufficiently detailed.

Validity of the findings

The expanded discussion now addresses all important findings of the analyses and puts them into context. Very good!

---

## Round 0.3 · accepted · Accept

Thank you for addressing all of the reviewers' concerns. Your manuscript is now ready for publication.